# Nitrogen Sources in Young Peach Trees in the Presence and Absence of *Paspalum notatum* Co-Cultivation

Betania Vahl de Paula [1,*], Danilo Eduardo Rozane [2], Eduardo Maciel Haitzmann dos Santos [2], Beatriz Baticini Vitto [1], Jacson Hindersmann [1], Luis Eduardo Correa Antunes [3], Gilberto Nava [3], Arcângelo Loss [4], George Wellington Bastos de Melo [5], Fernando Teixeira Nicoloso [1] and Gustavo Brunetto [1]

[1] Soil Science Department, Federal University of Santa Maria, Av. Roraima, 1000—Camobi, Santa Maria 97105-900, RS, Brazil
[2] Agronomy Department, São Paulo State University, Campus de Registro, Av. Nelson Brihi Badur, 430—Vila Tupi, Registro 11900-000, SP, Brazil
[3] Embrapa Clima Temperado, Highway BR-392, Km 78, 9th District, Monte Bonito, Pelotas 95701-008, RS, Brazil
[4] Center for Agricultural Sciences, Federal University of Santa Catarina, Admar Gonzaga Highway, 1346, Itacorubi, Florianopolis 88034-000, SC, Brazil
[5] Embrapa Uva e Vinho (Grape and Wine), Livramento Street, 515—Centro, Bento Gonçalves 95701-008, RS, Brazil
* Correspondence: behdepaula@hotmail.com

**Abstract:** Nitrogen (N) sources are applied to soils cultivated with peach trees. But, soil cover crops, as *Paspalum notatum*, a Pampa biome native species, commonly present in orchards, can absorb part of N, decreasing the amount used by peach trees. The study aimed to evaluate N absorption and physiological parameters of young peach trees cultivated in soil with the presence and absence of *Paspalum notatum*. The experiment was carried out for 180 days in a greenhouse, where N sources were applied to peach trees in the presence or absence of *Paspalum notatum*. Urea and organic compost were used. Dry matter, tissue N and physiological parameters were evaluated in peach trees. Dry matter and tissue N were evaluated in *Paspalum notatum*. Nitrogen in soil was evaluated. The N uptake by the peach trees with urea application, on average, was 32% higher than the N uptake by the peach trees in the control treatment or with organic compost, regardless of the presence or absence of *Paspalum notatum*. Cultivation with *Paspalum notatum* decreased N uptake by peach trees and, consequently, peach trees photosynthetic pigment content, and stimulated senescence anticipation in about 30 days. However, the total dry matter of peach trees cultivated with *Paspalum notatum* in any of the treatments applied was not modified. These results may guide new ways of co-cultivating cover crops and young peach trees.

**Keywords:** *Prunus persica* L.; cover crops; mineral source; organic source

## 1. Introduction

The most determining steps for successful orchard implantation, after choosing rootstock and peach cultivar (*Prunus persica*), are planting, growth and maintenance fertilization [1,2]. Pre-planting fertilization is carried out before transplantation of seedlings and it is usually recommended in order to raise phosphorus (P) and potassium (K) levels to ones above critical level [2,3]. However, N is generally not applied in pre-planting fertilization, and, when given, doses are small due to plants little root system development [3,4]. This decreases N uptake chances, which can easily have some lost forms, as nitrate ($NO_3^-$) in the leaching process. Therefore, N sources are usually applied in growth fertilization, and doses defined based on soil organic matter content [5].

Urea is the N source commonly applied on peach orchards, since, among other factors, it usually presents lower cost per N unit. In order to avoid root system damage, urea is applied without incorporation, and in plants crown projection, since it is considered

as one of the regions with the highest root volume [2,3]. However, urea solubilization and hydrolysis is very rapid, which can reduce the use of applied N by peach trees [6,7]. Therefore, an interesting strategy is the use of other N sources, as the organic compost, where N mineralization is more gradual, potentiating its use by the peach trees [8–10]. Thus, an increase in nutrients concentration inside the plant is expected, which can be diagnosed by foliar analysis and, or, photosynthetic rate as well as growth parameters, for example, dry matter production [11,12].

However, in peach orchards, such as those in South America, native cover crops species from the Pampa biome, such as *Paspalum notatum*, can be observed in lines and interlines, when the chosen option is not to desiccate with residual herbicides [13]. *Paspalum notatum* is a very important cover species due to its wide distribution and adaptability in Pampa biome natural pastures, which extend along the border between Brazil, Argentina and Uruguay. Favored by its fast-growing habit, together with its adaptability and good dry matter production, it has been considered one of the most promising native forage grasses in the region and its distribution is gradually increasing in fruit orchards, constituting a way to improve its preservation [14,15].

Orchard coverage plants, as *Paspalum notatum,* dissipate raindrops kinetic energy, reducing soil hydric erosion, especially in orchards as the ones located in Rio Grande do Sul, Brazil, in sandy soil or undulating relief [16–19]. On the other hand, *Paspalum notatum* can also absorb part of N contained in urea or organic compound, applied as N source in peach trees. This reduces N availability to the tree and may negatively affect its growth because of lower N uptake and photosynthetic rate [17,20]. However, the actual competition for N applied from different sources is not sufficiently known for *Paspalum notatum*, which could, in turn, contribute to decision making, for example, in whether or not to graze, desiccate or even maintain coverage plants present in orchards without intervention. Our hypothesis is that the young peach trees when co-cultivated with *Paspalum notatum* uptake more N from organic source. The study aimed to evaluate absorption and physiological parameters in young peach trees cultivates in soil together with *Paspalum notatum.*

## 2. Materials and Methods

### 2.1. Soil

Experiment was conducted along 180 days, in a greenhouse, in controlled temperature and humidity (25 °C mean temperature and 60% mean relative air humidity). Soil samples from Typic Hapludalf [21] were collected in 0–0.20 m layer. Soil was air dried, sieved in a 2.00 mm mesh and reserved. One sample was submitted to chemical analyses [22]. Phosphorus and K content were corrected by addition of 0.037 g P kg$^{-1}$ of soil and with 0.019 g K kg$^{-1}$ of soil. The doses were applied to increase the P and K contents until reaching the appropriate fertility range for peach cultivation in soils with medium clay content and high cation exchange capacity at pH 7.0 (CEC$_{pH\,7.0}$) [2,3].

### 2.2. Peach Tree and Treatments

Peach tree seedlings, cultivar Chimarrita, grafted on Capedebosq rootstock were transplanted to 8 kg pots containing 6.20 kg of soil, one plant per pot. Post-grafting seedlings were one year old and were produced from softwoood cuttings.

Before transplantation to pots containing peach tree, *Paspalum notatum* seedlings collected in natural field (29°43′34.55″ S e 53°45′30.47″ W geographical coordinates) washed in distilled water, selected and multiplied, were cultivated during 90 days in rectangular propylene boxes (0.60 × 0.60 × 0.10 m) containing sand. At every 30 min, irrigation with Hoagland nutritive solution at 25% its original concentration was performed [23].

At 15 days following peach trees transplantation, six 10 cm height *Paspalum notatum* seedlings, which is naturally occurring in orchards from Brazil Southern Region, were transplanted to pots belonging to *Paspalum notatum* consortium treatment.

At 21 days following coverage plants implantation, treatments at a complete randomized experimental design, factorial 2 × 3, with 5 repetitions were applied. *Paspalum notatum*

presence and absence was implanted in the following treatments: Control (no fertilization); urea fertilization and organic compost fertilization.

Administered urea had 45% total N and was applied as single dose before irrigation. Organic compost was produced from grape juice agroindustry residues, such as must and rachis, as well as poultry residues and sawdust, containing 20 g kg$^{-1}$ total N, 4 g kg$^{-1}$ ammonium (N-NH$_4{}^+$), 4 g kg$^{-1}$ nitrate (N-NO$_3{}^-$), 7.4 g kg$^{-1}$ total P, 24 g kg$^{-1}$ total K, 193 g kg$^{-1}$ total organic C, 432 g kg$^{-1}$ dry matter, pH 9.0 in water and 9.65 C/N relation as chemical composition. Urea and organic compost were applied in each pot surface, as 40 kg N ha$^1$ dose, which is indicated as adequate by the fertilization and liming manual for peach tree orchards implantation [5], which was equivalent to 0.28 g urea and 20.5 g organic compost per plant. For dose calculations, spacing between plants of 3 × 5 m and soil density of 1 g cm$^{-3}$ were considered.

At *Paspalum notatum* containing treatment, plant shoot was cut at 15 cm from soil surface, simulating mowing. About 25% of the waste from each pot was reserved for analysis and the remainder was added to the soil surface of each pot, simulating the mowing that takes place in commercial orchards.

Throughout the experiment the average temperature in the greenhouse was 25 °C and air humidity was 70%. Every two days the pots were weighed and, when necessary, distilled water was added to maintain the field capacity close to 60%.

### 2.3. Physiological Parameters

At 120 days following peach seedlings transplantation, three leaves from each repetition were collected, frozen in liquid N$_2$ and kept at −80 °C until further analysis. Leaves carotenoids, chlorophyll *a* (Chl *a*) and chlorophyll *b* (Chl *b*) content were evaluated [24]. Pigment concentration was obtained according to methodology proposed by Lichtenthaler [25].

Also, at 120 days following peach trees transplantation, gas exchange was also measured using a portable gas infrared analyzer (IRGA—Infra-red Gas Analyzer) with artificial red and blue light sources (LI-6400XT LI-COR, Inc., Lincoln, NE, USA). The CO$_2$ assimilation rates (µmol CO$_2$ m$^{-2}$ s$^{-1}$), water conductance (mol H$_2$O m$^{-2}$ s$^{-1}$), CO$_2$ intracellular concentration (µmol CO$_2$ mol$^{-1}$), transpiration rate (mmol H$_2$O m$^{-2}$ s$^{-1}$) and leaf temperature (°C) in all treatments were analyzed between 10 h to 12 h.

### 2.4. Dry Matter and N in the Soil and Tissue

At 180 days, soil samples from 0–0.20 m layer were collected from each pot. Soil was prepared and ammonium and nitrate extraction was performed using a 1 mol L$^{-1}$ KCl solution. The NH$_4{}^+$ and N-NO$_3{}^-$ contents in the soil were analyzed in the Kjeldahl semi-micro steam distillation distiller (Tecnal, TE-0363, Brazil). The N-mineral content was calculated according Tedesco et al. [22].

On the same date, the plants were cut close to the soil surface. The peach trees were separated into leaves and stem. All the mass of *Paspalum notatum* aerial part was collected. Roots were manually separated from the soil, washed in running water and then in distilled water. Organ samples were dried in a forced air circulation oven at 65 °C [13,22].

### 2.5. Statistical Analysis

Results were submitted to analysis of variance (ANOVA) with two factors (fertilizer source x *Paspalum notatum* absence or presence) and when treatments effect was significant, results were submitted to Tukey test (minimum significant difference, $p < 0.05$).

## 3. Results

The highest concentrations of nitrate and mineral N (nitrate + ammonium) were observed in soil containing urea and organic compost in the absence of *Paspalum notatum* (Figure 1) (Table S1). In soil grown in the absence of *Paspalum notatum* and with the application of urea, we observed 13.6% more mineral N in the soil, when compared to soil

grown in the absence of *Paspalum notatum*, but with the application of organic compost (Figure 1).

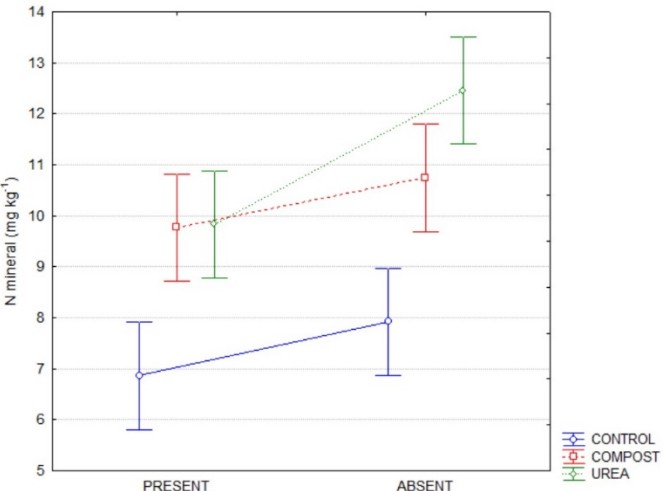

**Figure 1.** N mineral content in soil with peach trees and submitted to urea and organic compost application, after 180 days of cultivation, in *Paspalum notatum* presence and absence. Two-way ANOVA. Tukey test at 5% probability. The values presented are averages of three replicates per treatment and the vertical bar represents the standard error.

The N uptake in peach leaves was influenced by the interaction of the evaluated factors (Table S1). N uptake in peach leaves with urea application in the presence of *Paspalum notatum* did not differ from the other treatments. However, N uptake in peach leaves with urea application in the absence of *Paspalum notatum* was superior to control and organic compost (Figure 2A) (Table S1). N uptake in the stems of peach trees grown in soil in the presence or absence of *Paspalum notatum* showed no significant difference for the applied fertilizers (Table S1).

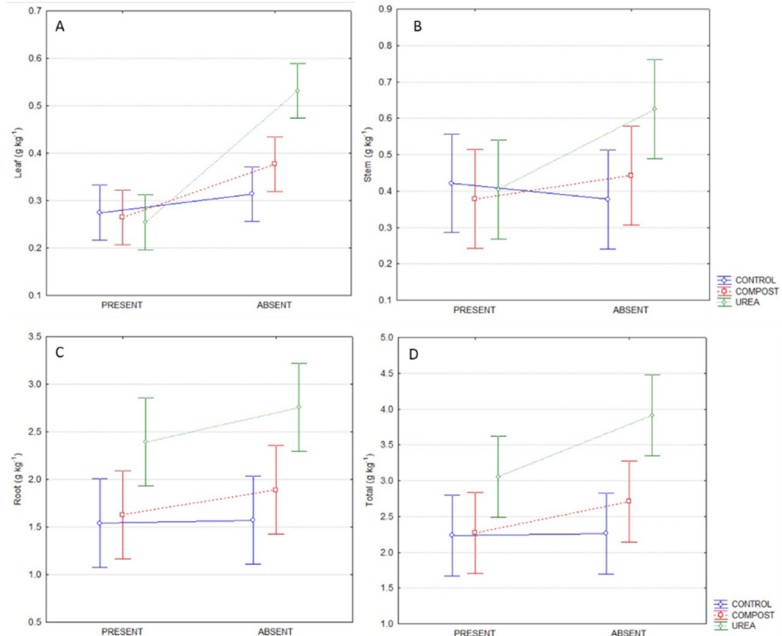

**Figure 2.** Nitrogen uptake in leaves (**A**), stems (**B**), roots (**C**) and total (**D**) from peach trees cultivated in pots with urea and organic compost application, in *Paspalum notatum* presence and absence. Two-way ANOVA. Tukey test at 5% probability. The values presented are averages of three replicates per treatment and the vertical bar represents the standard error.

On the other hand, N uptake in peach trees roots and N uptake in peach trees (Figure 2C,D) with urea application showed the highest N concentrations regardless of the presence or absence of *Paspalum notatum* (Table S1). The N uptake by the peach trees with urea application, on average, was 32% higher than the N uptake by the peach trees in the control treatment or with organic compost, regardless of the presence or absence of *Paspalum notatum* (Table S1).

The cultivated peach trees showed no significant difference in the production of leaf and stem dry matter in any of the treatments (Figure 3A) (Table S1). However, peach trees cultivated in soil with urea application showed higher production of root dry matter, when compared to the application of organic compost, in the presence and absence of *Paspalum notatum* (Table S1).

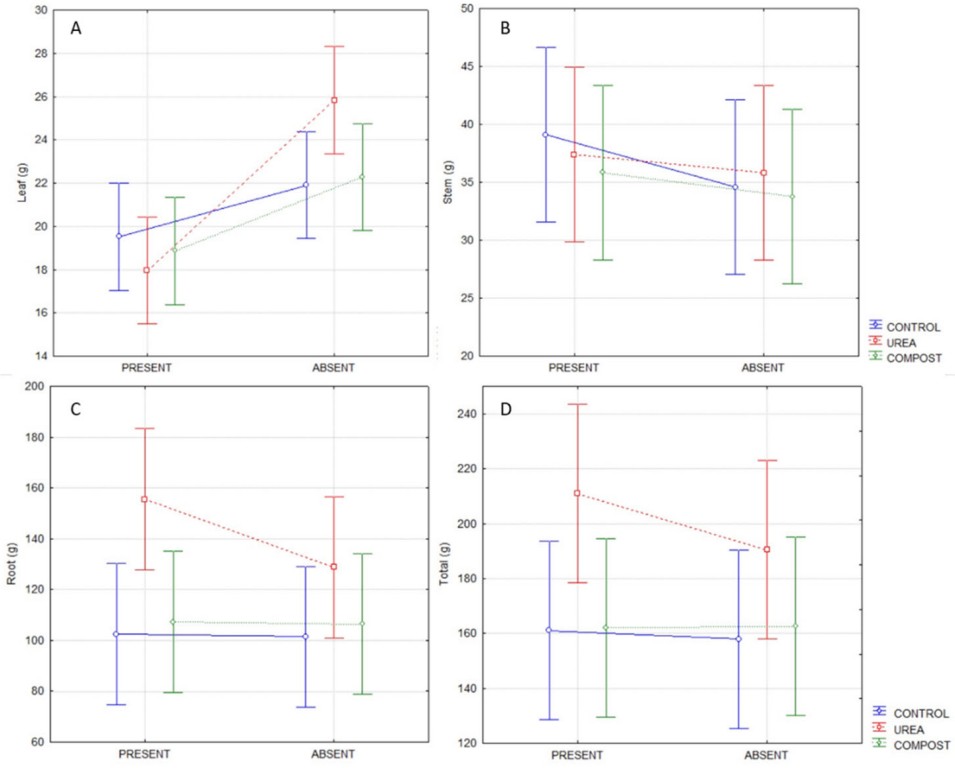

**Figure 3.** Dry matter of leaves (**A**), stem (**B**), roots (**C**) and total (**D**) of peach trees cultivated with urea and organic compost application, in *Paspalum notatum* presence and absence. Two-way ANOVA. Tukey test at 5% probability. The values presented are averages of three replicates per treatment and the vertical bar represents the standard error.

The production of shoot dry matter of *Paspalum notatum*, when intercropped with peach, was greater when urea was applied (Table S1). Furthermore, we observed a greater amount of N absorbed by peach trees with the application of urea, when compared to peach trees grown in the control soil and with the application of organic compost (Figure 4A,B). In the roots, the N absorbed by the root system did not differ statistically between N sources (Table S2), indicating that *Paspalum notatum* tends to accumulate more N in the shoot than in the roots, unlike what was observed in peach trees. The total dry mass production of *Paspalum notatum* grown in peach intercrop was higher with the application of urea (Figure 4) (Table S2).

The leaves of peach trees cultivated in the absence of *Paspalum notatum* in the control soil and with applications of urea and organic compost, showed the highest values of chlorophyll *a* (Figure 5A). The leaves of peach trees cultivated in the absence of *Paspalum notatum* with applications of urea, showed the highest values of chlorophyll *b* and carotenoids (Figure 5B,C). The values of photosynthetic rate and stomatal conductance did not differ

statistically between peach trees cultivated in the presence and absence of *Paspalum notatum*, in the control soil, with applications of urea and organic compost (Figure 5D,E).

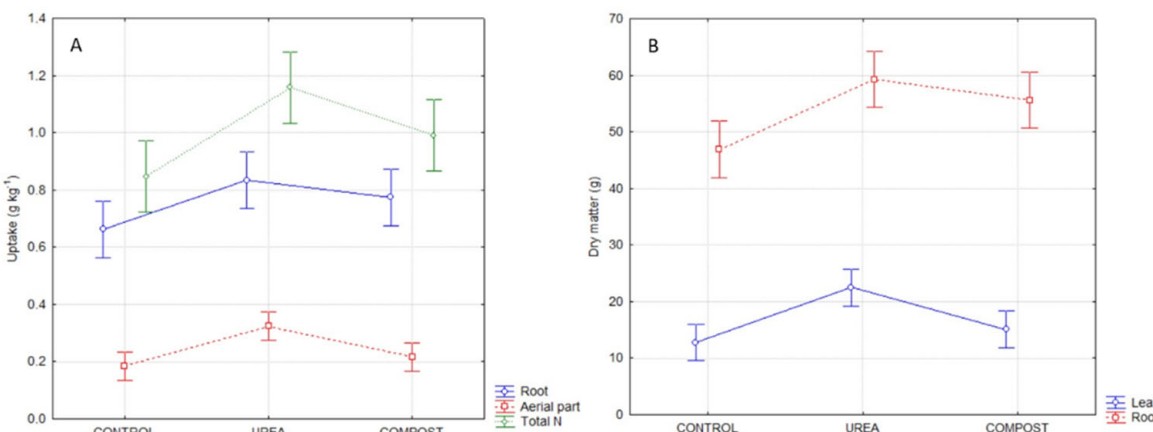

**Figure 4.** N uptake in aerial part, roots and total (**A**) and dry matter (**B**) of *Paspalum notatum* cultivated in intercropping with peach trees, with application of urea and organic compost. Two-way ANOVA. Tukey test at 5% probability. The values presented are averages of three replicates per treatment and the vertical bar represents the standard error.

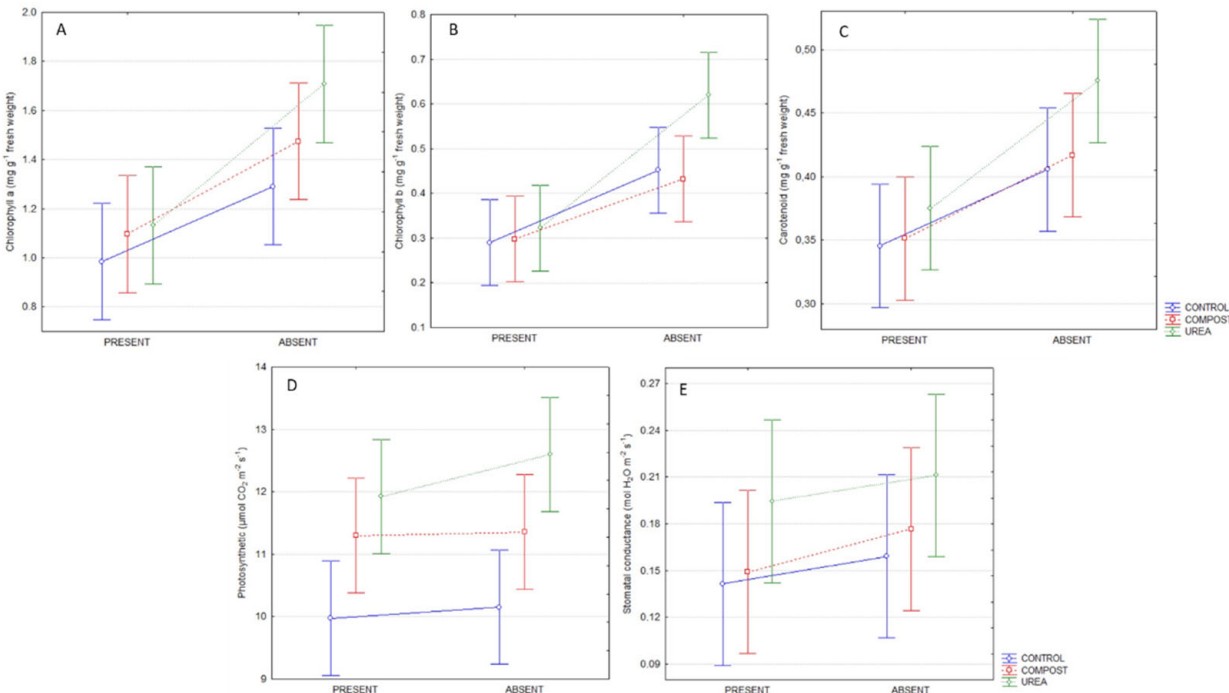

**Figure 5.** Chlorophyll *a* (**A**), chlorophyll *b* (**B**), carotenoids (**C**), photosynthesis (**D**) and conductance (**E**) concentrations in leaves from peach trees cultivated in pots with urea and organic compound application, *Paspalum notatum* presence and absence. Two-way ANOVA. Tukey test at 5% probability. The values presented are averages of three replicates per treatment and the vertical bar represents the standard error.

## 4. Discussion

The higher mineral N contents observed in the soil with urea and organic compost in the absence of *Paspalum notatum* can be explained by the high uptake of soil nutrients, including N, by the roots of *Paspalum notatum,* which they transport to the shoot, as observed by the N concentration and accumulation in the tissue.

The higher contents of different forms of N in the soil with urea application are explained by increased hydrolysis of urea by extracellular urease enzymes produced by soil microorganisms such as bacteria and fungi, producing ammonium carbonate $((NH_4^+)_2CO_3)$, which is not stable in the soil. When in contact with water, it decomposes into $HCO_3^-$, $OH^-$ and $NH_4^+$. $HCO_3^-$ can then decompose into $CO_2$ and $OH^-$. $NH_4^+$ reacts with $OH^-$, $NH_3$ can be transferred to the atmosphere. However, part of the remaining $NH_4^+$ can be transformed by biological oxidation into nitrite $(NO_2)$, followed by $NO_3^-$ [26]. Thus, part of the N forms can be taken up by cover plants, for example, also present in orchards and by fruit trees, such as peach trees, used in the present study [15]. This explains the interaction between the factors (fertilizer x *Paspalum notatum*) observed for $NH_4^+$ in the soil.

In soils without the presence of cover plants, part of the N forms, such as nitrate, can be lost, mainly by leaching, as observed by Oliveira et al. [17], in an apple orchard subjected to the application of urea and organic compost. On the other hand, the mineralization of organic residues in soil, such as organic compost is more gradual and, therefore, generally the increment of N forms in soil occurs more slowly over time [27]. This may account for the lower N contents observed in the potting soils. This may be desirable, as there may be a greater synchronization between N mineralization and uptake by the fruit trees, since roots tend to grow throughout the year [28,29] and also, therefore, can absorb N. The organic compost used in the present study presented a C/N ratio close to 20, in which a balance occurs between mineralization and immobilization of N [27,30], which is evidenced through the increase in N forms observed in the soil.

The greater amount of N absorbed, mainly by the leaves, in the absence of *Paspalum notatum* is a result of the greater availability of mineral N in the soil [6,31,32]. While in the roots, higher amounts of N absorbed were observed in peach trees cultivated in soil with urea application. The root is a reserve organ where much of the N can be accumulated in the form of proteins [7,31,32]. The proteins present in the roots can be degraded and part of the N can be redistributed in different ways to the growing organs, such as leaves and branches of the year [6]. Thus, the plant may be less dependent on N applied through different sources due to the accumulation of internal N reserves in the stem and roots [33].

There was no significant difference between the total dry mass of peach trees cultivated with and without *Paspalum notatum*. This demonstrates that intercropping with cover crops does not reduce the dry mass of the peach trees.

However, there was a difference between the sources of N. The peach trees cultivated with the application of urea obtained higher production of dry matter, possibly because the urea is more soluble than the organic compost. This may have contributed to a greater uptake of N, increasing the dry mass of the roots, especially in the absence of *Paspalum notatum*. Thus, the roots can exploit a greater volume of soil, increasing the absorption of nutrients, including N [34].

Even with the lower of mineral N, which also reduced the N absorbed by the leaves and the values of total photosyn stem, thetic pigments (for example, chlorophyll a, chlorophyll b and carotenoids) and leaf availability of the senescence stemmed peach trees, the total dry matter of peach trees grown with *Paspalum notatum* was not negatively affected. This is desirable because it enables ground cover plants, including *Paspalum notatum*, to be used in orchard rows. *Paspalum notatum* can contribute to the dissipation of the kinetic energy of the raindrop, which reduces the potential for water erosion, especially on soils located on undulating slopes. Also, the maintenance of ground cover plants is justified because part of the N applied, in the form of urea or organic compost, can be absorbed, remaining in the roots and shoot of the plants and, therefore, can return to the soil after organ senescence, reducing the amount of N lost to the environment and also increasing its potential use by peach trees later [35–37].

The highest production of total dry mass of *Paspalum notatum* was observed in peach intercropping, with the application of urea. This may have happened because of the rapid solubilization of urea, which increases the availability of N forms in the soil, some of which

may be taken up by *Paspalum notatum*, which may have a higher growth rate of roots and aboveground parts, compared to peach.

The higher values, for exemple, of chlorophyll *a* and chlorophyll *b* in leaves of peach trees grown in soil in the absence of *Paspalum notatum* can be explained by the greater availability of mineral N forms, including in soil with urea and organic compost applications. This increases the potential for N uptake by plants and increases N concentration in the tissue. Part of the absorbed N may be part of the chlorophyll molecule, being formed by the association between a central Mg atom (pheophytin complex) and four N atoms from four symmetric pyrrole rings [38]. The higher values of carotenoids, which were also observed in peach trees with urea application and without *Paspalum notatum*, can be explained because these compounds act in preventing photo-oxidative damage, caused by highly reactive oxygen species produced in photosynthesis [39,40].

We emphasize that the present study simulates the co-cultivation of *Paspalum notatum* and peach trees in the growing phase. At this stage, it is important to look for ways to maintain ground cover in orchards without harming peach tree growth. The results obtained can guide new ways of managing the co-cultivation of cover crops and peach trees in the growth phase.

## 5. Conclusions

Peach trees cultivated in soil with urea absorbed more N when compared to the application of organic compost, even in the presence of *Paspalum notatum*. The cultivation of peach trees associated with *Paspalum notatum* decreased the uptake of N and, consequently, decreased the photosynthetic pigment content of peach trees, as well as stimulated the anticipation of leaf senescence by 30 days. However, there was no reduction in the total dry matter production of young peach trees due to the cultivation of *Paspalum notatum*.

Therefore, during growth, young peach roots are concentrated in the topsoil and more soluble N sources, such as urea, can be absorbed more efficiently (Figure S1). On the other hand, the presence of cover crops near the roots of the peach trees in the growth phase may compete for the applied N. Thus, we suggest that during peach trees growth, cover crops such as *Paspalum notatum* are kept only between the rows of the orchard.

**Supplementary Materials:** The following supporting information can be downloaded at: https://www.mdpi.com/article/10.3390/agronomy12112669/s1.

**Author Contributions:** B.V.d.P., E.M.H.d.S., B.B.V. and J.H., collected data and metadata, organized the data for modelling, ran models, co-wrote the paper. D.E.R., G.N., G.W.B.d.M., F.T.N. and L.E.C.A. co-wrote the paper. A.L. and G.B. revised the paper. All authors have read and agreed to the published version of the manuscript.

**Funding:** We acknowledge the financial support of the Coordenação de Aperfeiçoamento de Pessoal de Nível Superior (CAPES), Conselho Nacional de Desenvolvimento Científico e Tecnológico (CNPq) and Fundação de Amparo à Pesquisa do Rio Grande do Sul (FAPERGS).

**Conflicts of Interest:** The authors declare no conflict to interest.

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
