# Peer review of "Nitrogen Sources in Young Peach Trees in the Presence and Absence of Paspalum notatum Co-Cultivation"

_agronomy, doi:10.3390/agronomy12112669_

Round 1
Reviewer 1 Report (New Reviewer)
Paula et al. have investigated the N absorption by peach trees in presence and absence of a cover crop “Paspalum notatum”. As per my opinion, this study is preliminary research in this field and doesn’t have so much novel significance with respect to the Agronomy journal. However, some of the flaws are enlisted below;
1. Line 27: Paspalum notatum should be always italic.
2. In abstract, authors have mentioned about morphological and physiological parameters of peach trees but they didn’t discuss the results with numerical evidences.
3. Authors could expand their study by investigating the molecular mechanism of N uptake by peach trees, but they studied very preliminary variables.
4. The first two sentences of Introduction lack citations.
5. First author has cited his/her 3 papers in the manuscript, including one paper which is not in English.
6. The discussion part is also needed to be improved.
7. I found 53% plagiarism in the manuscript file. Introduction and Methodology sections are almost fully copied.
Author Response
Line 27: Paspalum notatum should be always italic.
Reply: We accept your suggestion and adjust the text.
In abstract, authors have mentioned about morphological and physiological parameters of peach trees but they didn’t discuss the results with numerical evidences.
Reply: We accept your suggestion and that of another reviewer and adjust the text.
Authors could expand their study by investigating the molecular mechanism of N uptake by peach trees, but they studied very preliminary variables.
Reply: We appreciate the excellent suggestion. However, the study aimed to evaluate absorption and physiological parameters in young peach trees cultivated in soil together with Paspalum notatum.
We plan to write another manuscript with more data soon that will incorporate your suggestion. Thank you very much!
The first two sentences of Introduction lack citations.
Replay: We inserted references number 1, 2 and 3.
First author has cited his/her 3 papers in the manuscript, including one paper which is not in English.
Reply: The chapter cited which is not in English has been replaced.
The first author studied peach culture during her Master's and PhD. And she is a member of a research group with researchers from several countries that studies fruit tree fertilization.
The discussion part is also needed to be improved.
Reply: We accept your suggestions and the text was reviewed.
I found 53% plagiarism in the manuscript file. Introduction and Methodology sections are almost fully copied.
Reply: We accept your suggestion and the text was reviewed and modified.
Reviewer 2 Report (Previous Reviewer 1)
The current stady reports investigation to evaluate N absorption and physiological parameters of young peach trees cultivated in soil in the presence and absence of Paspalum notatum where the Nitrogen sources used were urea and organic compost. This is very important in terms of obtaining natural nitrogen sources and reducing N leaching while contributing to the long-term sustainability of agroecosystems. The research undertaken in this paper supplements and updates the knowledge on the optimization of nitrogen fertilization.
The article presented to me for review contains all the key elements required in the scientific and editorial work of the journal. I only have a few points that I suggest to the authors for considering when preparing a revised version.
Results
• The description of the results, line 244 should be consistent with the numbering in Figure 5 (5c, 5d, 5e)
Discussion
• In my opinion, referring to specific tables and graphs in the discussion is unnecessary, a detailed description was presented in the results chapter.
Conclusions
• Conclusions should include a comparison of the therapy and the potential benefits of choosing the best solution, as well as recommendations and prospects for the future. Meanwhile, they look more like a summary of the results.
Author Response
The current stady reports investigation to evaluate N absorption and physiological parameters of young peach trees cultivated in soil in the presence and absence of Paspalum notatum where the Nitrogen sources used were urea and organic compost. This is very important in terms of obtaining natural nitrogen sources and reducing N leaching while contributing to the long-term sustainability of agroecosystems. The research undertaken in this paper supplements and updates the knowledge on the optimization of nitrogen fertilization.
Reply: We appreciate your feedback and suggestions to improve the quality of the manuscript.
The article presented to me for review contains all the key elements required in the scientific and editorial work of the journal. I only have a few points that I suggest to the authors for considering when preparing a revised version:
Results
The description of the results, line 244 should be consistent with the numbering in Figure 5 (5c, 5d, 5e)
Reply: We adjust the figure and the text.
Discussion
In my opinion, referring to specific tables and graphs in the discussion is unnecessary, a detailed description was presented in the results chapter.
Reply: We accept your suggestion and remove references to specific tables and figures in the discussion.
Conclusions
Conclusions should include a comparison of the therapy and the potential benefits of choosing the best solution, as well as recommendations and prospects for the future. Meanwhile, they look more like a summary of the results.
Reply: We accept your suggestion and a paragraph was inserted with the information in the conclusions.
Reviewer 3 Report (New Reviewer)
This study investigated nitrogen uptake by peach trees with and without cover crops. Reducing nitrogen loss by planting with cover crop is one useful method and the theme of the study is within topic of the journal. the focus of the study is straightforward and the study includes some important findings. However, in my reading of the manuscript, there are some concerns about the description of the results and discussion. Major revisions are required in this manuscript.
Major comments
To reduce nitrogen loss after fertilizer application, N uptake by the total plant is very important. Although N uptake of peach trees and cover crops were shown, the sum of these plants were not presented. In my viewing the data, in the urea plots, total N uptake with the cover crop was less than that without it. Plant N uptake partly contribute to reduce nitrogen loss but actual nitrogen loss in the study is not clearly known. At least, the antagonistic effects of cover crops on nitrogen uptake of peach trees is significant.
The presentation of the data is not complete. There are many data but some information is missing. In table 1, data of N uptake is shown and this data would be same values in figure1. In the results, data about N concentration in table 1 is referred (L284-285), but there is no information. I suppose the data of N uptake in table 1 should be replaced by that of N concentration. In addition, standard deviation should be indicated. The results of statistical analysis are not complete. This would be just the interaction effects. The results of main effects also should be indicated.
The statistical analysis should be performed for the data in figure 2-5. The explanation for bars should be included.
Detail comments.
L21-22, I do not agree with this statement. The potential risk of nitrogen losses after urea application is more important.
L27, what does it mean by morphology? There is no data on morphology in the study.
L29, N concentration data is missing in this study.
L34-35, I do not understand this sentence. The data in figure 3 is not consistent with the results for N uptake in figure 2. N concentration data is required.
L253, figure 6 is not important. Effects of cover crops on N uptake differed between N fertilization treatment.
Author Response
To reduce nitrogen loss after fertilizer application, N uptake by the total plant is very important. Although N uptake of peach trees and cover crops were shown, the sum of these plants were not presented. In my viewing the data, in the urea plots, total N uptake with the cover crop was less than that without it. Plant N uptake partly contribute to reduce nitrogen loss but actual nitrogen loss in the study is not clearly known. At least, the antagonistic effects of cover crops on nitrogen uptake of peach trees is significant.
Reply: We appreciate the comment. We believe that part of the N may have been lost by volatilization. However, the losses were probably small, because the temperature inside the greenhouse was controlled. However, volatilization has not been evaluated in the present study because it was not the objective of the study.
The presentation of the data is not complete. There are many data but some information is missing. In table 1, data of N uptake is shown and this data would be same values in figure1. In the results, data about N concentration in table 1 is referred (L284-285), but there is no information. I suppose the data of N uptake in table 1 should be replaced by that of N concentration. In addition, standard deviation should be indicated. The results of statistical analysis are not complete. This would be just the interaction effects. The results of main effects also should be indicated.
Reply: We accept your suggestion and the statistic was inserted in all figures. We have transferred Table 1 to the supplement to avoid repetition in the presentation of data as cited by reviewer 3. Also, We inserted two more tables as a supplement, which presents all the data with statistics and interaction.
The statistical analysis should be performed for the data in figure 2-5. The explanation for bars should be included.
Reply: We accept your suggestion and statistical analysis was inserted in Figures 1-5. The explanation for the bars has been added.
Detail comments.
L21-22, I do not agree with this statement. The potential risk of nitrogen losses after urea application is more important.
Reply: The text has been revised and modified.
L27, what does it mean by morphology? There is no data on morphology in the study.
Reply: We accept your suggestion and that of another reviewer and adjust the text.
L29, N concentration data is missing in this study.
Reply: The data is in table S1.
L34-35, I do not understand this sentence. The data in figure 3 is not consistent with the results for N uptake in figure 2. N concentration data is required.
Reply: We adjust the text and insert the statistic into the figure. We also added tables as supplemental material. Thus, there was no statistical difference between the presence and absence of Paspalum notatum in the control treatment (without fertilization) in the total dry mass of the peach tree.
L253, figure 6 is not important. Effects of cover crops on N uptake differed between N fertilization treatment.
Reply: We accept your suggestion and the figure 6 was removed.
Reviewer 4 Report (New Reviewer)
The main goal of the study was to to evaluate absorption and physiological parameters in young peach trees cultivates in soil together with Paspalum notatum. The Authors carried out the two-factor greenhouse experiment along 180 days. The Authors showed interesting results and they suggest that the results obtained can guide new ways of managing the co-cultivation of cover crops and peach trees in the growth phase. Work written correctly, research methods selected correctly as well, results presented in a clear manner, enough literature included, conclusions answered on the main aims of the study.
In the text just a few corrections are needed (marked in the file).

Author Response
The main goal of the study was to to evaluate absorption and physiological parameters in young peach trees cultivates in soil together with Paspalum notatum. The Authors carried out the two-factor greenhouse experiment along 180 days. The Authors showed interesting results and they suggest that the results obtained can guide new ways of managing the co-cultivation of cover crops and peach trees in the growth phase. Work written correctly, research methods selected correctly as well, results presented in a clear manner, enough literature included, conclusions answered on the main aims of the study.
Reply: We appreciate your feedback and appreciate your suggestions to improve the quality of the manuscript.
In the text just a few corrections are needed (marked in the file):
Line 74: delete dot
Reply: We accept your suggestion and adjust the text.
Line 74: delete K and space
Reply: We accept your suggestion and the text was revised.
Line 191: space
Reply: We accept your suggestion and the text was revised.
Round 2
Reviewer 1 Report (New Reviewer)
The authors didn't revise their manuscript seriously.
- I asked them to write the abstract with numerical evidence, and they said they accepted my suggestion but didn't.
- Last time, I detected plagiarism 53%. Now, it is 50%. There is no significant difference.
Author Response
The authors didn't revise their manuscript seriously.
- I asked them to write the abstract with numerical evidence, and they said they accepted my suggestion but didn't.
Reply: The abstract has been revised and numerical evidence has been added (line 29-31).
- Last time, I detected plagiarism 53%. Now, it is 50%. There is no significant difference.
Reply: We have reviewed the manuscript. We scanned the manuscript in CopySpider software (https://copyspider.com.br). This software considers a 3% similarity threshold, which represents a similarity statistic of up to 3% that is not considered plagiarism. Our manuscript was scanned and the highest similarity rate found was 0.61%. The methodology text was also revised, because portions of the methodology in the present manuscript can be observed in other articles, for example:
Ambrosini, V.G., Voges, J.G., Benevenuto, R.F., Vilperte, V., Silveira, M.A., Brunetto, G., Ogliari, J.B., 2015. Single-head broccoli response to nitrogen application. Científica 43, 84. https://doi.org/10.15361/1984-5529.2015v43n1p84-92
Sete, P.B., Comin, J.J., Nara Ciotta, M., Almeida Salume, J., Thewes, F., Brackmann, A., Toselli, M., Nava, G., Rozane, D.E., Loss, A., Lourenzi, C.R., da Rosa Couto, R., Brunetto, G., 2019. Nitrogen fertilization affects yield and fruit quality in pear. Sci. Hortic. (Amsterdam). 258, 108782. https://doi.org/10.1016/j.scienta.2019.108782
Reviewer 3 Report (New Reviewer)
Dear authors
I reviewed the revised manuscript. Although the manuscript has improved substantially, there are some concerns as described below.
The data about N concentration is still missing. In the revised manuscript dry weight and nitrogen content were presented in figure 2 and 3. Nitrogen concentration is calculated from these values but data is still missing. Root dry weight with cover crops in the urea treatment was higher than that without cover crops. However, for nitrogen content in roots with cover crops was lower than that without cover crops. In this situation, there should be a great difference of nitrogen concentration between with and without cover crops. I suppose the situation is not usual and there would be some mistakes about the calculation. The data of nitrogen concentration should be added in tables or graphs. Moreover, there is no description in the results or discussion.
The results of statistical analysis are not revised. In the former review, I suggest to include the results of main effects; fertilizer effects and cover crop effects. In the present manuscript, only interaction effects are presented. This is not enough.
Author Response
Revisei o manuscrito revisado. Embora o manuscrito tenha melhorado substancialmente, existem algumas preocupações, conforme descrito abaixo.
The data about N concentration is still missing. In the revised manuscript dry weight and nitrogen content were presented in figure 2 and 3. Nitrogen concentration is calculated from these values but data is still missing. Root dry weight with cover crops in the urea treatment was higher than that without cover crops. However, for nitrogen content in roots with cover crops was lower than that without cover crops. In this situation, there should be a great difference of nitrogen concentration between with and without cover crops. I suppose the situation is not usual and there would be some mistakes about the calculation. The data of nitrogen concentration should be added in tables or graphs. Moreover, there is no description in the results or discussion.
The results of statistical analysis are not revised. In the former review, I suggest to include the results of main effects; fertilizer effects and cover crop effects. In the present manuscript, only interaction effects are presented. This is not enough.
Reply: We appreciate the comments. The results have been reviewed and corrections were made.
In order to avoid confusion, we chose to redo all figures and eliminate the use of percentage for N concentrations.
We redone all the statistical analysis and inserted in the supplement two tables with all variables with significance at 1% and 5%. The values for the interaction between the factors are also presented in the supplement (Table S1).
Added to this, we will make available the database obtained in excel.
Root dry weight with cover crops in the urea treatment was higher than that without cover crops. However, for nitrogen content in roots with cover crops was lower than that without cover crops. In this situation, there should be a great difference of nitrogen concentration between with and without cover crops. I suppose the situation is not usual and there would be some mistakes about the calculation. The data of nitrogen concentration should be added in tables or graphs. Moreover, there is no description in the results or discussion.
Reply: The results have been reviewed and the discussion has been improved.
Os resultados da análise estatística não são revisados. Na revisão anterior, sugiro incluir os resultados dos efeitos principais; efeitos de fertilizantes e efeitos de culturas de cobertura. No presente manuscrito, são apresentados apenas efeitos de interação. Isso não é o bastante.
Resposta: Agradecemos os comentários. Todos os resultados todos os resultados foram revisados e a estatística qualificada, conforme descrito acima.
This manuscript is a resubmission of an earlier submission. The following is a list of the peer review reports and author responses from that submission.
Round 1
Reviewer 1 Report
The manuscript entitled "Nitrogen sources in young peach trees in the presence and absence of Paspalum notatum co-cultivation” is a good study in order to explain N absorption and physiological parameters of peach trees cultivated in soil in the absence and presence of Paspalum notatum. This knowledge can be of great practical importance and contribute to deciding whether to use herbicides in the orchards, exposing themselves to contamination of the soil environment or maintaining cover plants without intervention, while limiting the losses of N.
The paper presented to me for review contains all the key elements required in the scientific work and editors of the journal. However, after going through the manuscript I have some suggestions for the improvement of the manuscript:
Abstract
- No future prospective and potential benefits of this research are provided at the end of the abstract. Please provide that in a single line.
Introduction
- Give hypothesis of study at the end of the introduction. Also, give the target audience who will benefit from this research.
Materials and Methods
- The cited entries in lines 120 and 126 are not included in the Reference section.
Results
- I suggest replacing the word abstract in line 167 with the word explanation.
- Please review the complete description of the results and correctly assign the numbers of figures and tables to the description of the text. I noticed that there are errors (examples: line174 is figure 2c and it should be 2d; line 206 is table 2 but such table does not exist). This also applies to the caption of Figure 4 as well as the letter markings in Figure 5.
Conclusions
Give potential benefits of a selection of best amendment. Conclusion statement with the recommendation and future prospective.
Author Response
The manuscript entitled "Nitrogen sources in young peach trees in the presence and absence of Paspalum notatum co-cultivation” is a good study in order to explain N absorption and physiological parameters of peach trees cultivated in soil in the absence and presence of Paspalum notatum. This knowledge can be of great practical importance and contribute to deciding whether to use herbicides in the orchards, exposing themselves to contamination of the soil environment or maintaining cover plants without intervention, while limiting the losses of N.
The paper presented to me for review contains all the key elements required in the scientific work and editors of the journal. However, after going through the manuscript I have some suggestions for the improvement of the manuscript:
Abstract
Nenhum benefício futuro e potencial desta pesquisa é fornecido no final do resumo. Forneça isso em uma única linha.
Resposta: Acatamos a sugestão e fizemos essa alteração no texto.
Introdução
Dê hipótese de estudo no final da introdução. Além disso, indique o público-alvo que se beneficiará com esta pesquisa.
Resposta: Acatamos a sugestão e fizemos essa alteração no texto.
Materiais e métodos
As entradas citadas nas linhas 120 e 126 não estão incluídas na seção Referência.
Resposta: Adicionamos no texto.
Resultados
Sugiro substituir a palavra resumo na linha 167 pela palavra explicação.
Resposta: Acatamos a sugestão e fizemos essa alteração no texto.
Por favor, revise a descrição completa dos resultados e atribua corretamente os números de figuras e tabelas à descrição do texto. Percebi que existem erros (exemplos: line174 é a figura 2c e deveria ser 2d; a linha 206 é a tabela 2 mas essa tabela não existe). Isso também se aplica à legenda da Figura 4, bem como às marcações de letras na Figura 5.
Resposta: Aceitamos a sugestão e fizemos essa alteração no texto
Conclusões
Dê os benefícios potenciais de uma seleção de melhor alteração. Declaração de conclusão com a recomendação e prospectiva futura.
Reviewer 2 Report
The study entitled "Nitrogen sources in young peach trees in the presence and absence of Paspalum notatum co-cultivation" aimed to evaluate N absorption and physiological parameters of young peach trees cultivated in soil in the presence and absence of Paspalum notatum is interesting.
Introduction, Material and methods, results and conclusion are appropriated and adequately descripted.
Author Response
The study entitled "Nitrogen sources in young peach trees in the presence and absence of Paspalum notatum co-cultivation" aimed to evaluate N absorption and physiological parameters of young peach trees cultivated in soil in the presence and absence of Paspalum notatum is interesting.
Introduction, Material and methods, results and conclusion are appropriated and adequately descripted.
Replay: We thank you for your review and availability.
Reviewer 3 Report
Dear Authors,
I have studied the manuscript and found it interesting. In general, it is focused on analysing the competition of functional plant cover crops to the main crop (peach trees) for nitrogen supply. I think the use of cover crops is important task especially for maintainance of soil health and biodiversity in agricultural systems. In following terms, the search of appropriate species with good characteristics (optimal coverage and competitivnes for weeds, but low direct influence for the main crop) is desired and each potentialy useble cover crop should be tested prior its recommandation for use in praxis.
In general the manuscript is almost good and clearly written. I have just few suggestions for improvements:
- I didn´t understood the phrase "high interpretation range" (line 85), as this seems to me not an obvious term for interpreting soil quality. Moreover the refference used is in Spain language, so I was not able to follow the methodology of the soil evaluation (but the reference is something not really seen as problem).
- "herbaceous cuttings... if we speak about perennial crops, like peach, their multiplication is done eather by hardwood or softwood cuttings (if we dont speak about in-vitro techniques). So better term would be "softwoood cuttings".
- I was wondering, how many samples were analyzed for nitrogen content in peach. Did you measured all plants or only those used for physiological measurements?
- I would recommend you to describe the section with "2.5. Statistical analyes" in more detail. For example, I didn´t found the information about the statistical intervals used for comparison of the treatments in the mentioned figures. Are these Sd or confidential intervals? Did you tested the data quality? In some treatments, there was quite large variability in the measured parameters.
- The total value of absorbed N (line 174) - is it mentioned in Figure 2c or 2d?
- The sentence "On the other hand, N taken up by roots did not differ statistically between soil in the presence and absence of Paspalum notatum (Table 1)." (line 174-175) - this is little bit tricky... as the roots forwards the N to all other organs, it is difficult to speak about uptake. You should interpret it rather as content in roots, eapecially because you do not quantify only the N taken up during the trial but the total N also comming from the previous accumulation in reserves. Moreover, in this sentence, I was not clear about this statement because of likely different results presented in Table 1 and Figure 2 for roots of peach grown with or without the cover crop.
- Finally I have one additional concern - you have done the study on young, non-bearing peaches. Do you expect no effect on the trees in the later development? In the results of your study, you have mentioned that the amount of nitrogen absorbed was lower with use of the P. notatum, you found lower chlorophyl pigments content, and you expect also higher leaf senscence rate. The study was done for 1 season and for one-year-old trees. The total growth of such young trees is usually dominated by the use of reserve N. Furthermore trees with fruit set have usually significantly higher photosynthetic rate, so the decrease in pigments could lead to unexpected limits for the trees productivity in the later development. The similar photosynthetic rate as well as the final biomass of the young trees can be therefore misleading. I suggest to the authors to elucidate the preliminarity of the results.
Good luck with your manuscript.
MM.
Author Response
I have studied the manuscript and found it interesting. In general, it is focused on analysing the competition of functional plant cover crops to the main crop (peach trees) for nitrogen supply. I think the use of cover crops is important task especially for maintainance of soil health and biodiversity in agricultural systems. In following terms, the search of appropriate species with good characteristics (optimal coverage and competitivnes for weeds, but low direct influence for the main crop) is desired and each potentialy useble cover crop should be tested prior its recommandation for use in praxis.
In general the manuscript is almost good and clearly written. I have just few suggestions for improvements:
I didn´t understood the phrase "high interpretation range" (line 85), as this seems to me not an obvious term for interpreting soil quality. Moreover the refference used is in Spain language, so I was not able to follow the methodology of the soil evaluation (but the reference is something not really seen as problem).
Replay: We've adjusted the sentence in the text to be clearer.
"herbaceous cuttings... if we speak about perennial crops, like peach, their multiplication is done eather by hardwood or softwood cuttings (if we dont speak about in-vitro techniques). So better term would be "softwoood cuttings".
Reply: We have accepted the suggestion and we have made this alteration in the text
I was wondering, how many samples were analyzed for nitrogen content in peach. Did you measured all plants or only those used for physiological measurements?
Reply: The N content was analyzed in all plants.
I would recommend you to describe the section with "2.5. Statistical analyes" in more detail. For example, I didn´t found the information about the statistical intervals used for comparison of the treatments in the mentioned figures. Are these Sd or confidential intervals? Did you tested the data quality? In some treatments, there was quite large variability in the measured parameters.
Reply: Yes. the data obtained were tested for normality.
The total value of absorbed N (line 174) - is it mentioned in Figure 2c or 2d?
Reply: 2d. We have made this alteration in the text.
The sentence "On the other hand, N taken up by roots did not differ statistically between soil in the presence and absence of Paspalum notatum (Table 1)." (line 174-175) - this is little bit tricky... as the roots forwards the N to all other organs, it is difficult to speak about uptake. You should interpret it rather as content in roots, eapecially because you do not quantify only the N taken up during the trial but the total N also comming from the previous accumulation in reserves. Moreover, in this sentence, I was not clear about this statement because of likely different results presented in Table 1 and Figure 2 for roots of peach grown with or without the cover crop.
Reply: We have accepted the suggestion and we have made this alteration in the text
Finally I have one additional concern - you have done the study on young, non-bearing peaches. Do you expect no effect on the trees in the later development? In the results of your study, you have mentioned that the amount of nitrogen absorbed was lower with use of the P. notatum, you found lower chlorophyl pigments content, and you expect also higher leaf senscence rate. The study was done for 1 season and for one-year-old trees. The total growth of such young trees is usually dominated by the use of reserve N. Furthermore trees with fruit set have usually significantly higher photosynthetic rate, so the decrease in pigments could lead to unexpected limits for the trees productivity in the later development. The similar photosynthetic rate as well as the final biomass of the young trees can be therefore misleading. I suggest to the authors to elucidate the preliminarity of the results.
Good luck with your manuscript. MM.
Reply: We understand your point. The peach tree is a crop that takes about 2-3 years to start production. And in this period of growth, many producers and agronomists do not know which sources of N to use, especially if cover crops are present. The present study showed how the plant behaves when the cover crop is grown close to the peach tree (in the planting row) and with different sources of N.
In the production phase, cover crops generally do not compete with peach trees because the roots of the peach tree tend to go deeper into the soil.
So to make it clear, we added a paragraph to the text that says that the results can be applied to peach orchards in the growing phase.
Thank you so much for your dedication and contribution!
Reviewer 4 Report
Manuscript ID.: HORTICULTURAE-1697217
Title: Nitrogen sources in young peach trees in the presence and absence of Paspalum notatum co-cultivation
Journal: Horticulturae. Special Issue “Strategies in Fruit Trees and Vegetables to Increase Nutrient Use Efficiency”
This manuscript compares the pot growth of peach trees in soils treated with two different nitrogen sources (urea and organic compost) in the presence or absence of Paspalium notatum as soil cover crops. The topic is particularly interesting in the part that deals with the problems related to the development of fruit trees in the presence of orchard cover crops such as Paspalium notatum, a very common cover crop species in South America. The goal is to see what the effects of this coexistence might be if the decision was made to maintain this native cover crop species in the orchard without interventions aimed at limiting its growth. The topic is well framed in the introduction but has significant gaps in the subsequent paragraphs.
The materials and methods must be modified because they are written in a superficial way and have several inaccuracies and require some clarifications. For example:
_ in section 2.1 the soil analyses should be described in more detail as the references given are not readily available and in any case not found on the English language web
_ in 2.2 the C/N parameter is dimensionless and cannot be indicated as a percent. The relative reported value for compost composition seems not to be real considering the reported values for total organic C and total N. Furthermore, taking it for granted that the compost composition values refer, as correctly they should, to the dry matrix, the quantities of nitrogen added in the single pots in the form of urea and/or compost do not seem to be equivalent: 0.289 g of urea (45% total N) correspond to 0.13 g total N; 19.95 g of organic compost (2% total N referred to dry compost) correspond to 0.399 total N if this quantity refers to dry weight) and/or 0.17 g total N if instead we are talking about wet compost.
_in 2.3, two bibliographic references are cited that are given loosely (authors' last name and year) and not identified with a number. Among other things, these references are not indicated in the list of the bibliography.
_ the description of the analytical determinations (the results of which are given in table 1) performed on Paspalium Notatum is completely missing as well as lacking more accurate information regarding the simulated mowing of the grass (why, how much residue was put into the pot, how many times is this operation repeated in 180 days)?
_why are measurements of physiological parameters carried out only at 120 days, while complete plant analyses are carried out at 180 days?
In general, the description of analytical methods should be reorganized in a more orderly and thorough manner trying to correct the many inaccuracies present. In addition, the authors should provide the complete list of chemicals and reagents with purity.
The Results chapter should also be improved both in terms of text (there are several errors in the interpretation of the data) and because some tables and figures should be adjusted. For example:
_in tab 1 in the first column should be inserted N-NH4+ and N-NO3- instead of NH4+ and NO3- and the legend should be corrected (abstract??)
_the description of fig.4 in the text and in the legend does not correspond exactly to the representation of the single figures.
_table 2 is completely missing
The paragraph 4 and 5 should be completely revised. Particularly, in discussion the results should be better interpreted and compared with other references reporting similar studies, while the conclusions should be expanded by giving greater emphasis to the results obtained having in mind the purpose of the work and highlighting the novelties and improvements made by this study with respect to the topic under consideration. Moreover, in the discussion there are statements that are different from what is written in the abstract: in line 32-33 it is stated "no reduction in paech trees total dry matter was observed in any treatment applied"; this sentence in the opposite of what is written in lines 262-266 "Peach trees cultived with urea and without Paspalium notatum showed a greater production of dry matter when compared to peach trees cultived with organic compost...."
Therefore, this article has several flaws in the way it has been developed despite the fact that the topic, however well framed, may be of interest. It must therefore be rejected in my opinion
Author Response
This manuscript compares the pot growth of peach trees in soils treated with two different nitrogen sources (urea and organic compost) in the presence or absence of Paspalium notatum as soil cover crops. The topic is particularly interesting in the part that deals with the problems related to the development of fruit trees in the presence of orchard cover crops such as Paspalium notatum, a very common cover crop species in South America. The goal is to see what the effects of this coexistence might be if the decision was made to maintain this native cover crop species in the orchard without interventions aimed at limiting its growth. The topic is well framed in the introduction but has significant gaps in the subsequent paragraphs.
The materials and methods must be modified because they are written in a superficial way and have several inaccuracies and require some clarifications. For example:
_ in section 2.1 the soil analyses should be described in more detail as the references given are not readily available and in any case not found on the English language web
Reply: We have accepted the suggestion and we added a summary of the methods used.
_ in 2.2 the C/N parameter is dimensionless and cannot be indicated as a percent. The relative reported value for compost composition seems not to be real considering the reported values for total organic C and total N. Furthermore, taking it for granted that the compost composition values refer, as correctly they should, to the dry matrix, the quantities of nitrogen added in the single pots in the form of urea and/or compost do not seem to be equivalent: 0.289 g of urea (45% total N) correspond to 0.13 g total N; 19.95 g of organic compost (2% total N referred to dry compost) correspond to 0.399 total N if this quantity refers to dry weight) and/or 0.17 g total N if instead we are talking about wet compost.
Reply: Urea and organic compost were applied in each pot surface, as 40 kg N ha1 dose.
_in 2.3, two bibliographic references are cited that are given loosely (authors' last name and year) and not identified with a number. Among other things, these references are not indicated in the list of the bibliography.
Reply: We have accepted the suggestion and we added in text.
_ the description of the analytical determinations (the results of which are given in table 1) performed on Paspalium Notatum is completely missing as well as lacking more accurate information regarding the simulated mowing of the grass (why, how much residue was put into the pot, how many times is this operation repeated in 180 days)?
Reply: Tissue analyzes performed on peach and paspalum notatum followed the same method described. Plants were cut once in 180 days and all residue was placed on the ground, simulating what actually occurs in young orchards.
In general, the description of analytical methods should be reorganized in a more orderly and thorough manner trying to correct the many inaccuracies present. In addition, the authors should provide the complete list of chemicals and reagents with purity.
The Results chapter should also be improved both in terms of text (there are several errors in the interpretation of the data) and because some tables and figures should be adjusted. For example:
_in tab 1 in the first column should be inserted N-NH4+ and N-NO3- instead of NH4+ and NO3- and the legend should be corrected (abstract??)
Reply: We have accepted the suggestion and we have made this alteration in the text
_the description of fig.4 in the text and in the legend does not correspond exactly to the representation of the single figures.
Reply: We have made this alteration in the text
_table 2 is completely missing
Reply: We have made this alteration in the text
The paragraph 4 and 5 should be completely revised. Particularly, in discussion the results should be better interpreted and compared with other references reporting similar studies, while the conclusions should be expanded by giving greater emphasis to the results obtained having in mind the purpose of the work and highlighting the novelties and improvements made by this study with respect to the topic under consideration. Moreover, in the discussion there are statements that are different from what is written in the abstract: in line 32-33 it is stated "no reduction in paech trees total dry matter was observed in any treatment applied"; this sentence in the opposite of what is written in lines 262-266 "Peach trees cultived with urea and without Paspalium notatum showed a greater production of dry matter when compared to peach trees cultived with organic compost...."
Reply: In fact, in the abstract it was evident that the cultivation of paspalum notatum did not impact any of the sources. That is, when comparing the same source with and without paspalum notatum, there was no difference in the dry matter of peach trees. There is only difference between the sources (urea x compound). We have modified the abstract and quoted text to improve understanding.
Round 2
Reviewer 4 Report
The responses and changes made by the authors in the second version of the article did not lead to substantial improvements.
The authors only partially accepted the comments addressed to them, and many others were ignored altogether or adapted inaccurately. Below are timely responses (in red) to the authors' comments attached to the second version of the submitted article:
_ in section 2.1 the soil analyses should be described in more detail as the references given are not readily available and in any case not found on the English language web
Reply: We have accepted the suggestion and we added a summary of the methods used.
The description of the methods is somewhat imprecise in the terms used: moreover, it should have been included as early as section 2.1 and not at 2.4. In the same paragraph the determination of total nitrogen in different tissue samples is also described more extensively, as required. The method reported is the Kjeldhal method but again there are some inaccuracies. However in no description the authors provide the complete list of chemicals and reagents with brand name and purity as requested.
Note that the nitrogen determined by the Kjeldhal method does not represent as erroneously stated N-tot but rather organic N + N-NH4: to obtain total nitrogen one must add N-NO3.
_ in 2.2 the C/N parameter is dimensionless and cannot be indicated as a percent. The relative reported value for compost composition seems not to be real considering the reported values for total organic C and total N. Furthermore, taking it for granted that the compost composition values refer, as correctly they should, to the dry matrix, the quantities of nitrogen added in the single pots in the form of urea and/or compost do not seem to be equivalent: 0.289 g of urea (45% total N) correspond to 0.13 g total N; 19.95 g of organic compost (2% total N referred to dry compost) correspond to 0.399 total N if this quantity refers to dry weight) and/or 0.17 g total N if instead we are talking about wet compost
Reply: Urea and organic compost were applied in each pot surface, as 40 kg N ha1 dose.
Again, the response does not take into account the reviewer's comments
_in 2.3, two bibliographic references are cited that are given loosely (authors' last name and year) and not identified with a number. Among other things, these references are not indicated in the list of the bibliography.
Reply: We have accepted the suggestion and we added in text.
OK
_ the description of the analytical determinations (the results of which are given in table 1) performed on Paspalium Notatum is completely missing as well as lacking more accurate information regarding the simulated mowing of the grass (why, how much residue was put into the pot, how many times is this operation repeated in 180 days)?
Reply: Tissue analyzes performed on peach and paspalum notatum followed the same method described. Plants were cut once in 180 days and all residue was placed on the ground, simulating what actually occurs in young orchards.
However, the text continues to lack a sentence referring to analytical methods on Paspalum notatum. In addition, lines 115-117 state that "In the treatment with Paspalum notatum, plant shoots were cut 15 cm from the soil surface, simulating mowing. Some of the residue was reserved and the rest was added to the soil surface of the pots." This contradicts what was stated in the response, "......days and all residues were deposited on the soil, simulating what actually happens in young orchards."
In general, the description of analytical methods should be reorganized in a more orderly and thorough manner trying to correct the many inaccuracies present. In addition, the authors should provide the complete list of chemicals and reagents with purity.
The Results chapter should also be improved both in terms of text (there are several errors in the interpretation of the data) and because some tables and figures should be adjusted. For example:
_in tab 1 in the first column should be inserted N-NH4+ and N-NO3- instead of NH4+ and NO3- and the legend should be corrected (abstract??)
Reply: We have accepted the suggestion and we have made this alteration in the text
OK
_the description of fig.4 in the text and in the legend does not correspond exactly to the representation of the single figures.
Reply: We have made this alteration in the text
In the figure 4 legend still there is no correspondence between figure and description: Dry matter of shoot roots and total biomass are represented in 4a,4c,and 4e respectively and not as indicated 4a,4b 4c. Of course, the references for Total N uptake are also wrong (b,d,f no c,d,e)
_table 2 is completely missing
Reply: We have made this alteration in the text
The authors have removed the wording Table 2 in the text. Therefore, the parameter values related to photosynthetic activity, while very important, are not given in absolute value and tabular form at all.
The paragraph 4 and 5 should be completely revised. Particularly, in discussion the results should be better interpreted and compared with other references reporting similar studies, while the conclusions should be expanded by giving greater emphasis to the results obtained having in mind the purpose of the work and highlighting the novelties and improvements made by this study with respect to the topic under consideration. Moreover, in the discussion there are statements that are different from what is written in the abstract: in line 32-33 it is stated "no reduction in paech trees total dry matter was observed in any treatment applied"; this sentence in the opposite of what is written in lines 262-266 "Peach trees cultived with urea and without Paspalium notatum showed a greater production of dry matter when compared to peach trees cultived with organic compost...."
Reply: In fact, in the abstract it was evident that the cultivation of paspalum notatum did not impact any of the sources. That is, when comparing the same source with and without paspalum notatum, there was no difference in the dry matter of peach trees. There is only difference between the sources (urea x compound). We have modified the abstract and quoted text to improve understanding.
The authors, contrary to their request, made no changes to paragraphs 4 and 5 (discussion and conclusion). Therefore, the underlined remark was not considered by the authors.
Author Response
We do not agree with all of this reviewer's comments because some of their questions do not make sense. Therefore, we believe that he is not an expert in the area. We accept all Editor's revisions.
